# Hepatitis B Virus Knowledge and HBV-Related Surveillance Status Among HBsAg-Positive Patients in Qidong City: A Rural-Based Cross-Sectional Survey

**DOI:** 10.3390/healthcare13010017

**Published:** 2024-12-25

**Authors:** Hailiang Liu, Jing Hong, Zhaoxian Yan, Mei Li, Xiaofeng Zhai, Bo Pan, Changquan Ling

**Affiliations:** 1School of Integrative Medicine, Shanghai University of Traditional Chinese Medicine, Shanghai 201203, China; 2Department of Integrative Oncology, The First Affiliated Hospital of Naval Medical University, Shanghai 200433, China; 3School of Traditional Chinese Medicine, Naval Medical University, Shanghai 200433, China; 4Department of Integration of Chinese and Western Medicine, Peking University Cancer Hospital & Institute, Beijing 100142, China; 5The Fourth People’s Hospital of Jinan, Jinan 250102, China

**Keywords:** hepatitis B virus, knowledge, HBsAg-positive cohorts, multivariate logistic regression, cross-sectional survey

## Abstract

Objective: This study aimed to investigate hepatitis B knowledge and hepatitis B virus (HBV)-related surveillance status among HBsAg-positive patients, as well as to further explore the relevant influencing factors. Methods: A cross-sectional study was conducted on the HBsAg-positive patients from 8 October 2023 to 10 November 2023 in Qidong City. A self-report questionnaire was developed based on a literature review of similar studies. Univariate analysis of variance, multivariate logistic regression, and *t*-test analysis were conducted to analyze the collected data. Results: Of the 982 respondents who completed the on-site questionnaire, all participants were HBsAg-positive patients. Moreover, 51.32% had “good” knowledge of HBV. Multivariate logistic regression analysis showed that participants with a doctor in the family, those with an average monthly income above CNY 3000, and those with an average monthly income of CNY 1500–3000 were more likely to obtain a “good” cognitive evaluation (*p* < 0.001). The scores of the populations using HBV-related surveillance methods were low (2.02 ± 0.87); 64.87% (637/982) of the populations monitored had a score of no more than 2. Conclusions: This study suggests that the awareness of HBV prevention and treatment among participants, especially those of low-income classes and individuals lacking physician clinical management, should be promoted to increase the dissemination of HBV knowledge.

## 1. Background

HBV infection causes acute and chronic hepatitis, seriously jeopardizing public health, and about 350 million people worldwide are chronically infected with HBV [1]. According to a relevant study [1], about 3.61% of the global population is infected with HBV chronically, and in some countries and regions, the prevalence is more than 20% [2]. If not controlled, HBV infection can lead to chronic inflammation, liver fibrosis, and then cirrhosis, which can ultimately develop into hepatocellular carcinoma. In China, about 84 percent of liver cancer cases develop gradually from HBV infection. Therefore, there is a close relationship between chronic HBV infection and the onset of cirrhosis and hepatocellular carcinoma, which poses a major threat to the population’s health [3]. Though it is a major public health problem, and most countries and regions are very conscious of this health situation, only 10% of the 257 million HBV-infected individuals have been effectively diagnosed, among which only 1% have received adequate treatment [4]. Therefore, it is especially important to raise the level of proper public awareness about HBV.

The current studies on public perceptions of HBV have focused on cognitive differences among different populations and geographic regions, as well as relevant factors affecting cognitive differences [5,6]. A study in Pakistan showed that 88% of women had heard of HBV; however, only 34.8% of participants knew how to avoid contracting HBV, and 11.3% had received HBV-related surveillance in the year prior to the survey [7]. A survey on knowledge of hepatitis B in Nigeria showed that, while 77.8% of participants knew the name of HBV, only 47% had basic knowledge of hepatitis B. In a survey conducted in Gansu, China, only 1.5% of participants knew the name of HBV [8]. A survey in Gansu, China, showed that out of 400 cases of hepatitis B, 52% of the population had received a liver ultrasound and only 7% had been tested for HBV-DNA. In addition, an outpatient survey in China showed that only 25.62% of the hepatitis B population had received more than three types of hepatitis B monitoring [9]. Currently, there is no survey on HBV-related knowledge in the HBsAg-positive populations. Qidong City of Jiangsu Province, China, is a high-incidence area of liver cancer [10]. A community-based prospective cohort (Qidong hepatitis B infection cohort (QBC)) was established in 1996, and the present study team conducted a long-term follow-up of this cohort [11]. This study conducted a survey of HBsAg-positive patients in Qidong City, China, to gain insight into their level of HBV-related awareness and to further explore the factors influencing it. Conducting surveys on HBV-related knowledge among the public, especially among HBsAg-positive individuals, can shed light on the public’s knowledge of HBV. These findings can further provide data support for the development of public health policies and promote health education, which will make the HBsAg-positive individuals more willing to comply with the doctor’s treatment program and improve the effectiveness of treatment.

## 2. Materials and Methods

### 2.1. Study Design and Participants

The present study was conducted via an on-site paper questionnaire survey of HBsAg-positive cohorts who were part of the Early Diagnosis and Early Screening of Liver Cancer cohort conducted by the Institute of Liver Cancer Prevention and Control in Qidong City, Jiangsu Province, China, from 9 October 2023 to 10 November 2023. Those with intellectual and cognitive disabilities were excluded from the survey.

### 2.2. Ethical Approval

The study was approved by the Ethics Committee of the First Affiliated Hospital of Naval Medical University (No. CCHEC2013-119), and all methods and procedures were conducted in accordance with the Declaration of Helsinki. Verbal informed consent was obtained from all respondents, who were made aware of the purpose of the study, the research process, personally identifiable information and other personal information, and their rights in relation to participation in the survey.

### 2.3. Questionnaire

A self-administered questionnaire written in Chinese was used for the survey, and in some cases, the questionnaire was filled out on behalf of the participants based on their answers after the researcher had stated the questions. This was to assess the respondents’ knowledge of the HBV. The questionnaire was modified based on the literature review of similar studies [12,13,14,15,16,17]. The questionnaire consisted of three main sections. (1) Basic information about the participants, including demographic characteristics (name, gender, age, average monthly income, occupation, current smoking status, current drinking status, and the presence of a physician in the family). (2) Knowledge about HBV, its preventive measures and transmission routes, and the hazards of the disease. This section contained ten questions that aimed to evaluate whether the respondents’ HBV-related knowledge was good or poor based on the rate of answering the questions correctly. Scores of 60% and above were considered good knowledge, and scores lower than 60% were considered poor knowledge [18,19]. The relevant factors that affected the good or poor knowledge evaluation were analyzed. (3) Respondents’ HBV-related regular monitoring modalities, including alpha-fetoprotein (AFP), abdominal ultrasound, and liver function, with one point for each type of monitoring.

### 2.4. Pilot Study

Before the official start of the survey, we conducted a pre-survey of 80 HBsAg-positive individuals in Qidong City, China, to evaluate the knowledge score of the population through HBV-related questions. This was intended to determine the HBV knowledge of the pre-surveyed population and to learn about the population’s status regarding HBV-related surveillance. We did not encounter any problems during the pre-survey.

### 2.5. Sample Size Estimation

Based on a previous study [20], the probability of perceived adequacy of HBV was assumed to be 36% in the sample size calculation using the following formula:N = Z^2^_1 − α/2_P(1 − P)/e^2^
where N is the minimum number of patients required, Z^2^ is the 95% confidence interval (CI) of 1.96^2^, p is the estimated utilization rate, and e is 4% of the required accuracy. The estimated non-response rate was 5%, and the minimum sample size was 581 participants.

### 2.6. Data Analysis

SPSS 25.0 (Chicago, IL, USA) statistical software was used for data analysis. The socio-demographic information of the respondents was analyzed descriptively, and the results were expressed as whole numbers and percentages. We used univariate analysis of variance to analyze the relationship between socio-demographic information and cognitive appraisal of the respondents. Variables with *p* < 0.05 in univariate analysis were subjected to multivariate logistic regression to determine the independent predictors affecting cognitive well-being. Odds ratios (ORs) and 95% CI were used to describe these variables, with *p* < 0.05 considered a statistically significant difference.

## 3. Results

### 3.1. Participants’ Characteristics and HBV Cognitive Analysis

A total of 1022 individuals participated in this survey, of which 33 respondents submitted incomplete questionnaires and 7 respondents could not cooperate in completing the questionnaire. Therefore, the total effective recovery rate of the questionnaire was 96.10%, and 982 respondents were included in the inclusion analysis. The cognitive evaluation was determined based on the response scores of ten HBV-related questions.

Of all respondents, 54.99% were male and 45.01% were female. The median age of the participants was 65 (60, 69) years. The income levels were divided into three categories: below CNY 1500 per month, CNY 1500–3000 per month, and above CNY 3000 per month, and the percentages of participants in each category were 11.51%, 66.70%, and 21.79%, respectively. Moreover, 82.08% of the participants were farmers by occupation, 6.92% had a doctor in their household, 27.49% were current alcohol consumers, and 17.41% were smokers.

Based on the univariate analysis results, presence of a doctor in the family and monthly revenue were associated with HBV cognitive appraisal (*p* < 0.05) (Table 1).

The different answers given by the respondents to the ten questions related to HBV were used to clarify the respondents’ knowledge of HBV-related information, such as preventive measures, transmission routes, and the danger of the disease. We found that 66.70% of the respondents knew about HBV, 58.15% knew that HBV infection may develop into cirrhosis and liver cancer, 65.27% knew that HBV can be transmitted from mother to child, and 70.06% believed that HBV infection requires standardized treatment. Moreover, 50.71% believed that contact with hepatitis B patients could transmit HBV, and 47.35% believed that mosquito bites could transmit HBV (Table 2).

### 3.2. HBV-Related Monitoring Methods

The participants were asked about regular HBV-monitoring methods, including liver function, AFP test, and abdominal ultrasound. The respondents who understood and took the initiative to perform any monitoring methods were given a score of 1. It was found that the scores of the populations regarding HBV-related surveillance methods were low (2.02 ± 0.87); 64.87% (637/982) of the respondents monitored had scores of no more than 2, indicating they have relatively inadequate forms of daily active monitoring of their HBV status (Table 4).

This study found that the HBV-related surveillance score of those with adequate HBV-related knowledge was 2.39 ± 0.87 points, and the HBV-related surveillance score of those with inadequate HBV-related knowledge was 1.63 ± 0.86 points (Table 5). Spearman correlation analysis was used to test the correlation between the participants’ HBV-related knowledge ratings and regular surveillance scores, which showed a positive correlation (r = 0.434, *p* < 0.001). This suggests that the participants who know enough about HBV are more concerned about their condition and, therefore, have richer and more varied methods of regular surveillance.

## 4. Discussions

The current survey showed that the HBV knowledge of 51.3% of the respondents was rated as “good”. In a 2016–2017 survey on HBV and hepatitis C virus (HCV) awareness among residents of Chengdu and Chongqing, China, only 36.1% of the respondents had adequate knowledge of the two viruses [20]. Moreover, a 2010 survey conducted in Hong Kong to assess respondents’ understanding and knowledge of HBV infection showed that 53% did not know the clinical manifestations of acute hepatitis B [21]. A 2014 survey on knowledge of hepatitis B transmission in Guangdong, China, showed that 53.3% of respondents were unaware that HBV could be transmitted through unprotected sexual activity [22]. Although the HBV knowledge of the respondent group in the present survey is somewhat better than that reported in the two previous studies, the population’s knowledge of hepatitis B is still relatively inadequate. The two previous studies investigated the general population, while the current investigation consisted of all the HBsAg-positive patients. This may be because Qidong City is a high-prevalence area of hepatitis B and liver cancer in China, with a well-established local HBV screening system, and the individuals with HBV-related cases in this have more contact with medical personnel. Additionally, HBsAg-positive patients are more concerned about their health and have better HBV-related knowledge.

The participants of this study were from the Qidong hepatitis B infection cohort, QBC. The QBC population was mainly included in the elderly hepatitis B residents in the rural areas of Qidong City, which were tracked and followed up by our team for a long period of time, and the age distribution of this cohort was mainly concentrated in the age range of 50–70 years old, so 76.58% of the participants were over the age of 60 years old in this study. According to the research findings of the Qidong Liver Cancer Prevention and Control Institute, the age of liver cancer incidence in Qidong City showed a general increasing trend from 1972 to 2021, and the average age of liver cancer incidence in Qidong City in 2021 was 68.60 years old [23]. In China, the majority of hepatocellular carcinomas are transformed by HBV infection, so screening and follow-up are mainly carried out for the rural HBsAg-positive population in old age; thus, early diagnosis and treatment of hepatocellular carcinoma can be achieved. Our survey found that about half of the respondents believed that HBV could be transmitted by contact with hepatitis B patients and mosquito bites, suggesting that the respondents’ knowledge of non-transmission routes of HBV is low. Related surveys in countries such as Sudan [24], Malaysia [25], India [26], and Uganda [27] have shown that some groups of respondents also believe that HBV can be transmitted through eating together, mosquito bites, and sharing utensils. The poor knowledge of non-transmission routes and ambiguous or incorrect knowledge of HBV transmission routes may cause panic and discrimination against HBV-positive individuals. A survey on HBV stigmatization in Beijing, China [28], showed that HBV-negative people had higher stigma towards patients with chronic hepatitis B. Fear of HBV transmission may be the main cause of negative attitudes, which validates our scientific hypothesis that misperceptions about HBV transmission routes lead to panic and discrimination among the population.

The multivariate logistic regression analysis showed that income level influenced the knowledge of HBV, and individuals with higher income levels were more likely to have better HBV knowledge. Related studies in Nigeria [29], Turkey [30], Pakistan [31], and Congo [32] have also shown that the level of HBV knowledge among the population is higher in economically developed areas and urban areas than in rural areas, and that the economy level affects the prevalence of hepatitis B to some extent. A study in Turkey suggested that people with higher incomes were relatively less likely to be infected with HBV. African immigrants in the United States [33,34] and some immigrants in Thailand [35] had high rates of HBV infection due to factors such as low economic status and limited knowledge of HBV, resulting in low rates of hepatitis B screening. According to a previous study [36], income level directly impacts the living conditions of the population and their access to healthcare services. Higher-income groups can acquire more health-improving measures such as health insurance. To a large extent, low-income groups have less access to healthcare information, limiting their access to HBV knowledge. The economic status is one of the main challenges hindering the awareness and treatment of hepatitis B in high-prevalence areas and low- and middle-income countries [4], making it difficult to overcome HBV-related challenges globally. Therefore, there is a need to set up an effective and free screening system for HBV-related diseases and to increase the investment in medical resources. Only in this way can we better protect the high-risk groups and elderly infected patients from HBV-related liver diseases and reduce the incidence of cirrhosis and liver cancer.

This study also showed that people with a doctor in their family tended to know more about HBV, which may be due to the fact that HBsAg-positive patients have frequent contact or communication with doctors in their families, and thus obtain more direct information about the treatment and recovery process. A survey of patients with liver disease in Kuala Lumpur showed that the vast majority (93.6%) of patients with liver disease informed their family members about their condition [25], so these populations would be more proactive in seeking help if there were a doctor in the family. A survey in Turkey of clinical pharmacists’ educational interventions for hepatitis B patients showed that, after clinical pharmacists educated hepatitis B patients about HBV, hepatitis B patients’ knowledge about HBV and correct responses to HBV-related questions increased [37], which suggests that physicians and pharmacists have an important role in influencing the population to improve their knowledge and attitudes about HBV infection. A survey in the United States based on an Asian population showed higher levels of HBV screening among those who had a doctor’s recommendation or a family member’s recommendation [38]. It is, therefore, not difficult to understand that individuals with a physician among their family members have a higher level of knowledge and familiarity with HBV infection-related risk factors. Doctors among family members can help HBV-infected people in the following ways: providing common treatment options for HBV infection; instructing them to carry out regular monitoring of liver function, abdominal ultrasound, AFP, and other related monitoring; advising other members of the family to get vaccinated against Hepatitis B; listening to the infected person’s emotions and voices; encouraging them to engage in a healthy life; and even prescribing, if necessary, HBV-infected people with antiviral drugs and timely adjusting their treatment programs [39].

Notably, this study found that 64.87% (637/982) of participants had a score of no more than 2 for daily HBV-related monitoring practices, suggesting that the population is relatively deficient in terms of HBV infection-related monitoring practices. Moreover, individuals with adequate HBV-related knowledge pay more attention to their HBV infection status on a daily basis and adopt more monitoring methods. In a survey of medical personnel based on the Cambodian public health sector, more than 67% of the participants indicated that all HBV and HCV-infected patients should receive regular monitoring and education [40]. It is well known that regular follow-up of HBsAg-positive patients to assess disease progression with the help of AFP tests, liver function tests, and abdominal ultrasound is clinically important for the prevention of cirrhosis and hepatocellular carcinoma. The clinical value of abdominal ultrasound is that it can evaluate the structural changes of the liver, such as liver cirrhosis and liver tumors [41]. AFP is a sensitive indicator of liver cancer, and as its serum marker, the abnormal increase in AFP value is helpful for the early detection of liver cancer [42]. Liver function tests help us to understand the functional status of the liver and evaluate the degree of liver injury [43]. These examinations can help monitor the progress of hepatitis B, find complications, and guide treatment [44]. Therefore, it is necessary and socially important to help the population of HBsAg-positive cohorts to conduct regular surveillance and to form a social policy.

### Limitations

There are several limitations to this study. First, this study had a cross-sectional design based on a self-administered questionnaire; therefore, causality could not be directly inferred. Thus, further longitudinal studies are needed to validate possible causal relationships. Second, the questionnaire was adapted from multiple questionnaires, which could not be validated with a large sample size due to geographical limitations. Third, the participants in this study were mostly rural elderly adults, with a relatively low proportion of younger adults. Fourth, since our survey was conducted in Qidong City of Jiangsu Province, which is a high-prevalence area for hepatitis B and liver cancer in China, it does not fully represent the HBsAg-positive population’s knowledge of HBV and related antiviral status in Jiangsu or even nationwide. Fifth, with the introduction of treatment policies and the popularization of HBV knowledge, the proportion of the population with HBV knowledge may change in the future.

## 5. Conclusions

This study found that the overall knowledge of HBV among the interviewed population was insufficient. The presence of a doctor in the family and economic status affected the population’s knowledge of HBV. We recommend strengthening the awareness of HBV prevention and control among people with HBV-related conditions, especially low-income individuals and populations lacking clinical management by physicians, to increase the popularization rate of HBV knowledge for the HBsAg-positive population. In addition, we strongly recommend the development of a social policy for the HBsAg-positive population that can be used to help them monitor their infections periodically. This could effectively prevent HBV infection and related diseases.

## Figures and Tables

**Table 1 healthcare-13-00017-t001:** Socio-demographic information of participants.

Item	All Participants (N = 982)	Knowledge Good (N = 504)	Knowledge Poor (N = 478)	*p*-Value
Gender				0.235
Male	540 (54.99%)	290 (53.70%)	250 (46.30%)	
Female	442 (45.01%)	214 (48.42%)	228 (51.58%)	
Age (years old)				0.141
below 60	230 (23.42%)	116 (50.43%)	114 (49.57%)	
60–70	558 (56.82%)	284 (50.90%)	274 (49.10%)	
above 70	194 (19.76%)	104 (53.61%)	90 (46.39%)	
Occupation				0.112
Not farmers	176 (17.92%)	92 (52.27%)	84 (47.73%)	
Farmers	806 (82.08%)	412 (51.12%)	394 (48.88%)	
Current smoking status				0.526
No	811 (82.59%)	420 (51.79%)	391 (48.21%)	
Yes	171 (17.41%)	84 (49.12%)	87 (50.88%)	
Current drinking status				0.951
No	712 (72.51%)	365 (51.26%)	347 (48.74%)	
Yes	270 (27.49%)	139 (51.48%)	131 (48.52%)	
Presence of a doctor in the family				<0.001
No	914 (93.08%)	458 (50.11%)	456 (49.89%)	
Yes	68 (6.92%)	46 (67.65%)	22 (32.35%)	
Average monthly income (CNY)				<0.001
below 1500	113 (11.51%)	40 (35.40%)	73 (64.60%)	
1500–3000	655 (66.70%)	272 (41.53%)	383 (58.47%)	
above 3000	214 (21.79%)	192 (89.72%)	22 (10.28%)	

Data are presented as numbers (percentages). *p*-values were calculated via univariate analysis between the “knowledge good” and “knowledge poor” groups. CNY: China Yuan.

**Table 2 healthcare-13-00017-t002:** Respondents’ responses to HBV-related cognitive profiles (N = 982).

	Content of the Issue Item	Yes	No
1	Do you know about HBV?	655 (66.70%)	327 (33.30%)
2	Does HBV infection require standardized treatment?	688 (70.06%)	294 (29.94%)
3	Can chronic hepatitis B progress to cirrhosis and liver cancer?	571 (58.15%)	411 (41.85%)
4	Is vaccination the most effective way to prevent hepatitis B?	485 (49.39%)	497 (50.61%)
5	Can HBV be transmitted from mother to child?	641 (65.27%)	341 (34.73%)
6	Can HBV be transmitted sexually?	602 (61.30%)	380 (38.70%)
7	Can HBV be transmitted by blood?	590 (60.08%)	392 (39.92%)
8	Can HBV be transmitted by way of mosquito bites?	465 (47.35%)	517 (52.65%)
9	Can HBV be transmitted by contact with a person with hepatitis B?	498 (50.71%)	484 (49.29%)
10	Does hepatitis B vaccination prevent HBV infection?	643 (65.48%)	339 (34.52%)

Multivariate logistic regression revealed that the participants with a doctor in the family (OR = 2.029, 95% CI: 1.163–3.538, *p* < 0.001 vs. participants without a doctor in the family), those with an average monthly income above CNY 3000 (OR = 12.077, 95% CI: 7.018–20.782, *p* < 0.001 vs. below CNY 1500) or an average monthly income of CNY 1500–3000 (OR = 2.371, 95% CI: 1.516–3.707, *p* < 0.001 vs. below CNY 1500) were more likely to obtain a “good” cognitive evaluation (Table 3).

**Table 3 healthcare-13-00017-t003:** Multivariate logistic regression related to cognitive appraisal.

Item	Knowledge Good (N = 504)	Knowledge Poor (N = 478)	OR (95% CI)	*p*-Value
Presence of a doctor in the family				
No	458 (50.11%)	456 (49.89%)	/	Ref
Yes	46 (67.65%)	22 (32.35%)	2.029 (1.163–3.538)	<0.001
Average monthly income (CNY)				
below 1500	40 (35.40%)	73 (64.60%)	/	Ref
1500–3000	272 (41.53%)	383 (58.47%)	2.371 (1.516–3.707)	<0.001
above 3000	192 (89.72%)	22 (10.28%)	12.077 (7.018–20.782)	<0.001

*p*-values indicate whether the adjusted OR of a particular sub-category is significant compared to the reference category. OR: odds ratio; CI: confidence interval. CNY: China Yuan.

**Table 4 healthcare-13-00017-t004:** HBV surveillance scores of the participants (N = 982).

Surveillance Scores	0 Points	1 Point	2 Points	3 Points
numbers	33	257	347	345
percentages	3.36%	26.17%	35.34%	35.13%

**Table 5 healthcare-13-00017-t005:** The relationship between monitoring scores and HBV-related knowledge.

Surveillance Scores	0 Points	1 Point	2 Points	3 Points
knowledge rated as “good”	0.40% (2/504)	10.71% (54/504)	38.29% (193/504)	50.60% (255/504)
knowledge rated as “not good”	6.49% (31/478)	42.47% (203/478)	32.22% (154/478)	18.83% (90/478)

## Data Availability

The dataset supporting the conclusions of this article can be made available from the corresponding author upon reasonable request.

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
