# Peer review of "Hepatitis B Virus Knowledge and HBV-Related Surveillance Status Among HBsAg-Positive Patients in Qidong City: A Rural-Based Cross-Sectional Survey"

_healthcare, 2024, doi:10.3390/healthcare13010017_

Round 1
Reviewer 1 Report
Comments and Suggestions for Authors
Liu et al. conducted a cross-sectional study to investigate the HBV knowledge among HBsAg-positive patients in Qidong. The aim of this study is clear and it is well-designed. However, some issues should be dealt with.
1. Full names should be provided when abbreviations first appear. For example, HBV in Abtract.
2. Proofreading is suggested. Typos are found everywhere.
3. More information about HBV knowledge and HBV-related surveillance status should be provided in Introduction.
4. Adjust Table 1, the items are misaligned.
5. Did different groups of HBV-related knowledge have significantly different surveillance scores?
Comments on the Quality of English LanguageLanguage editing is suggested.
Author Response
Comments 1: Full names should be provided when abbreviations first appear. For example, HBV in Abtract.
Response 1: We thank the reviewers for their prompting and apologise for the omission, but have now added the full name of the abbreviation where it first appeared, and the full name of‘HBV in Abtract’ has been added (line 21 of the article), and all abbreviations throughout the article have been checked individually.
Comments 2: Proofreading is suggested. Typos are found everywhere.
Response 2: Thanks to the reviewers for their valuable suggestions on the language and expression of the article. We tried our best to improve the manuscript and made some changes in the manuscript. This paper has been handed over to a professional organisation for academic embellishment services to further embellish the language and relevant formatting of the article, thereby improving the quality of expression of this paper, and the relevant proofs are appended herewith. We appreciate for Reviewer’ warm workearnestly and hope that the correction will meet with approval.
Comments 3: More information about HBV knowledge and HBV-related surveillance status should be provided in Introduction.
Response 3: Thanks to the valuable suggestions of the reviewers, in the Background section of the article, new references have been added on the basis of the original one, which introduces the surveys on the knowledge of HBV and related surveillance status between China and Nigeria and Pakistan in the past 3 years, aiming at understanding the public's knowledge of hepatitis B, clarifying the information on HBV-related surveillance, and then grasping the current status of public awareness of HBV. The results of these studies can further provide data to support the development of public health policies, promote health education, and make HBsAg-positive people more willing to comply with the doctor's treatment plan and improve the effectiveness of the treatment (lines 54-63 of the article, and new references 8-9).
Comments 4: Adjust Table 1, the items are misaligned.
Response 4: We are very grateful to the reviewers for their careful review and for their hints about the contents of Table 1. We apologise for the error. Table 1 has now been adjusted so that the items in Table 1 are aligned, as shown in line 145 of the article content.
Comments 5: Did different groups of HBV-related knowledge have significantly different surveillance scores?
Response 5: We feel great thanks for your professional review work on our article. Participants were categorised into two groups with ‘good’ and ‘poor’ knowledge of hepatitis B based on their response scores to questions about hepatitis B. Subsequently, participants in both groups were asked about their daily monitoring, and those with good knowledge of HBV were given a score of (2.39 ± 0.87) and those with poor knowledge of HBV were given a score of (1.63 ± 0.86). Spearman's correlation analysis was used to test the correlation between participants' HBV-related knowledge scores and regular monitoring scores, and the results showed a positive correlation with a statistically significant difference (r = 0.434, P < 0.001), as detailed in lines 173-180 of the article.
Reviewer 2 Report
Comments and Suggestions for Authors
In the submitted manuscript “Hepatitis B Virus Knowledge and HBV-Related Surveillance Status Among HBsAg-Positive Patients in Qidong City: A Rural-Based Cross-Sectional Survey”, this study presents hepatitis B knowledge, and HBV-related surveillance status among HBsAg-positive patients in Qidong. The authors found that 51.32% of participants had "good" knowledge of HBV but HBV-related surveillance methods score of the populations was low. They recommend strengthening the awareness of HBV prevention and control among the people with HBV-related conditions, especially the low-income individuals and populations lacking clinical management by physicians, to increase the popularization rate of HBV knowledge for HBsAg-positive population.
However, there are some points that need to be clarified:
1. In the introduction, the authors should clarify why they focus on HBsAg positive patients. What is the importance of this population?
2. line 93-95, “scores of 60% and above were considered good knowledge, and scores less than 60% were considered poor knowledge”, why the authors select 60 as a cut-off? Please provide the references.
3. Table 1, what are the criteria to categorize the age? The authors mentioned “below 60” that is quite broad. The participants in this study were mostly elderly adults (76.58% were more than 60) with a relatively low proportion of younger adults. The median age should be presented. The authors should discuss more about the age of participants.
4. In section 3.2, the sentence in line 165-167 and the sentence in line 170-172 are quite similar, what is the difference?
5. HBV surveillance score of the participants (Figure 1) should be presented in table form with number and percentage of participants in each score.
Author Response
Comments 1: In the introduction, the authors should clarify why they focus on HBsAg positive patients. What is the importance of this population?
Response 1: Thank you very much for your detailed review of the manuscript. HBV infection causes acute and chronic hepatitis, which is a serious public health risk, and about 350 million people worldwide are chronically infected with HBV. According to relevant studies, the prevalence of HBV infection is more than 20 per cent in some countries and regions. If left unchecked, HBV infection leads to chronic inflammation, liver fibrosis, cirrhosis and eventually hepatocellular carcinoma. In China, about 84% of liver cancer cases are caused by the gradual development of HBV infection, which poses a major threat to the health of the population. This is a major public health problem, and most countries and regions are very concerned about this health condition, so it is especially important to raise the public's correct understanding of HBV. Currently, there is no HBV-related knowledge survey for HBsAg-positive people.
Qidong City, Jiangsu Province, China, is an area with a high prevalence of hepatocellular carcinoma. A prospective community-based cohort (Qidong hepatitis B infection cohort, QBC) was established in 1996, and this cohort has been followed up by our research team for a long period of time. In this study, HBsAg-positive patients in Qidong City, China, were surveyed to understand their level of knowledge related to HBV and to further explore the factors influencing it. Surveying the public, especially HBsAg-positive people, about HBV-related knowledge may provide insights into the public's awareness of HBV. The results of these studies can further provide data support for the development of public health policies, promote health education, make HBsAg-positive people more willing to comply with doctors' treatment plans, and improve the effectiveness of treatment. We think this is a good proposal. We have explained the research study context in further detail in the background section, see lines 39-45, 65-74 of the article.
Comments 2: line 93-95, “scores of 60% and above were considered good knowledge, and scores less than 60% were considered poor knowledge”, why the authors select 60 as a cut-off? Please provide the references.
Response 2:We take your suggestions very seriously. We have carefully reviewed the relevant literature and have added to the literature referenced for the scoring of this study. Referring to the HBV knowledge surveys conducted in recent years by Wang Fusheng in China, and Angga Dwiartama in Indonesia on 800 participants, a total score of >60% correct answers to hepatitis B-related questions was defined as ‘good knowledge’. The Hepatitis B knowledge section of this study consisted of 10 questions and the participants' knowledge of HBV was assessed according to the percentage of correct answers to the questions. A score of 60% and above was considered good knowledge and below 60% was considered poor knowledge. Relevant factors affecting the assessment of good and bad knowledge were also analysed. See lines 100-102 of the article and new references 18 (PMID: 28116112) and 19 (PMID: 35457514).
Comments 3: Table 1, what are the criteria to categorize the age? The authors mentioned “below 60” that is quite broad. The participants in this study were mostly elderly adults (76.58% were more than 60) with a relatively low proportion of younger adults. The median age should be presented. The authors should discuss more about the age of participants.
Response 3:The participant population of this study was from the Qidong hepatitis B infection cohort-QBC, the QBC population was mainly included in the hepatitis B residents in the rural areas of Qidong City. At that time, many participants were about 40 years old when they were first included in the cohort study. Due to the long-term follow-up by our team, 76.58% of the participants were over 60 years old at the time of this study in 2023. With the promotion and vaccination of hepatitis B vaccine in China and the continuous improvement of early screening and intervention measures for liver cancer, the incidence of hepatitis B and liver cancer in young people has gradually decreased. Therefore, few young people are included in this queue. According to the findings of the Qidong Institute of Liver Cancer Prevention and Control, the age of liver cancer incidence in Qidong City showed a general increasing trend from 1972-2021, and the average age of liver cancer incidence in Qidong City in 2021 was 68.60 years old (new reference #23). In China, most hepatocellular carcinomas are transformed by HBV infection, so screening and follow-up are mainly carried out for the HBsAg-positive population in the rural older age group, which will help to realize the early diagnosis and treatment of hepatocellular carcinoma.
The median age of the participants was 65 (60, 69) years, see lines 136-137 of this paper.
According to the suggestion of the reviewer, the age distribution characteristics of the participants in this study were described in the Discussion section, in order to more clearly describe the information characteristics of the geographical area and age of the HBsAg-positive population of this study species, which are detailed in the Discussion section, lines 201-212 of this paper.
Comments 4: In section 3.2, the sentence in line 165-167 and the sentence in line 170-172 are quite similar, what is the difference?
Response 4: Thanks to the reviewers for their detailed review and prompting of this article, these two paragraphs describe the surveillance scores of the ‘good’ and ‘poor’ groups of people with ‘good’ and ‘poor’ knowledge of hepatitis B. It was found that the HBV-related surveillance scores of those with sufficient and insufficient knowledge of HBV were ( 2.39 ± 0.87) and (1.63 ± 0.86) respectively. This is a duplicate description of the scores, so the two paragraphs were combined into one, see lines 173-175 of the article.
Comments 5: HBV surveillance score of the participants (Figure 1) should be presented in table form with number and percentage of participants in each score.
Response 5: Many thanks to the advice of the reviewers, Figure 1 here has been changed to Table 4. HBV surveillance scores of the participants. The table gives a specific description of the number and specific proportions of the different surveillance scores, see line 181 of the article.
Reviewer 3 Report
Comments and Suggestions for Authors
In this manuscript, Liu and coworkers report the outcome of a knowledge survey about hepatitis B conducted on 982 HBsAg(+) patients living in the Qidong district of the Jiangsu province in China. Qidong is considered as the place with the highest hepatocellular carcinoma incidence in China for a long time.
The authors observed that around 51% of the respondents have a good knowledge about hepatitis B and that the persons counting a medical doctor in their family or with an average monthly income above 1500 yuans obtained better cognitive scores that the others. The knowledge about surveillance methods of hepatitis B was mediocre with around 65% of the participants scoring low in that section of the questionnaire. The authors concluded that large efforts remain to be done to promote adequately the knowledge about HBV in China.
The paper is well-written, clear, and conclusions are sound.
There are few points that deserve further clarification.
The most important one concern the question “Do you know about HBV?”
First, it sounds more as a biological questions for students in University than as a medical question.
Why not asking “Do you know about hepatitis B?”?
Stemming from this observation, we notice that one third of the respondants do not know about HBV although they are themselves chronically infected. However, these participants, although they do not know answered anyhow to the next questions. What is the value of their answer? Did they answered by chance? What could be the value of the answers in persons who do not know about HBV? It is crucial that the authors comment this situation.
We do not know how many patients were treated. Hence, it is likely that those are more aware about the surveillance methods. Do the authors have an idea about the proportion of treated patients?
The authors do not asked any question about aflatoxin B1 that has been recognized to play a major role in liver tumorigenesis in Qidong. Why ? Questions about tobacco or alcohol are less “burning” that those about aflatoxin.
Line 268. The sentence about the self- administration of the questionnaire and causality is not easy to understand could you please explain with more details?
Author Response
Comments 1: The most important one concern the question “Do you know about HBV?”First, it sounds more as a biological questions for students in University than as a medical question. Why not asking “Do you know about hepatitis B?”?
Stemming from this observation, we notice that one third of the respondants do not know about HBV although they are themselves chronically infected. However, these participants, although they do not know answered anyhow to the next questions. What is the value of their answer? Did they answered by chance? What could be the value of the answers in persons who do not know about HBV? It is crucial that the authors comment this situation.
Response 1: We thank the reviewers for their interest in our team's study and their recognition of our research work. A prospective community-based cohort (Qidong hepatitis B infection cohort (QBC)) was established in Qidong, Jiangsu Province, which is a high prevalence area for hepatitis B in China, in 1996. Because this cohort had already been followed up and followed up for a long period of time by our research team, and we had also educated them about HBV-related knowledge in the process, the participants had a certain stock of HBV-related knowledge and could answer the questions in the questionnaire.
When we conducted this study, we wanted to know how much knowledge the participants had about HBV, whether they had a clear understanding of HBV, and to describe the extent of their understanding. Therefore, the question asked to the participants was ‘Do you know a lot about HBV?’, and I'm very sorry for the translation error, there was some misunderstanding. In the next study, if we change the question to ‘Do you know a lot about HBV?’(in Chinese ‘你是否很了解乙肝病毒?’), it will be more clear and intuitive, and we will continue to optimise and improve the questionnaire content, so that the results of the study will have a more effective guiding value. Thank you again for your constructive suggestions!
Comments 2: We do not know how many patients were treated. Hence, it is likely that those are more aware about the surveillance methods. Do the authors have an idea about the proportion of treated patients?
Response 2: We think this is an excellent suggestion. As the population in this study cohort were all HBsAg positive, some of these participants had received some previous treatment for Hepatitis B, some were currently on treatment, and of course a very small percentage of participants had never received treatment. Since this study is a real-world study, it was not convenient to categorise the treatment status of the participants in many of the above situations, so the study did not categorise whether or not they received treatment.
In response to the valuable suggestions and constructive comments made by the reviewers, it is indeed likely that the hepatitis B population who have received prior treatment therapy would know more about hepatitis B related monitoring methods. Therefore, in the next study, the research team will classify the participants into groups according to their previous, current and whether they have received hepatitis B treatment, and then carry out correlation analyses to continuously improve the scientific validity and guiding value of the results of the study, so as to better educate and guide HBsAg-positive people in the future, and to prevent or reduce the occurrence of malignant diseases such as hepatocellular carcinoma.
Comments 3: The authors do not asked any question about aflatoxin B1 that has been recognized to play a major role in liver tumorigenesis in Qidong. Why ? Questions about tobacco or alcohol are less “burning” that those about aflatoxin.
Response 3:Thank you very much for your professional comments on our article. When analysing the baseline information of the Qidong Hepatitis B Infection Cohort (QBC) population, information on alcohol consumption and smoking were included. Aflatoxin B1 is mostly found in mouldy grains, and due to the improvement of living standards in rural areas of QBC, mouldy grains are seldom consumed in rural areas of QBC, so the analysis of factors related to mouldy grains and aflatoxin was not included in the design. In response to the valuable comments of the reviewing experts, in the next while study, relevant factors such as aflatoxin B1 (mouldy grains), and other dietary habits will be included in the information of the study, with a view to continuously enriching the content and depth of the investigation.
Commets4: Line 268. The sentence about the self- administration of the questionnaire and causality is not easy to understand could you please explain with more details?
Response4: Thank you very much for the professional review of the reviewers. The subjects of the QBC were HBV infection related people, and we observed the transformation process from hepatitis B to liver cancer. More than 50% of the new cases of liver cancer in the world come from China every year(PMID: 25921660), and HBV infection accounts for 86.0% of liver cancer cases in China(PMID: 28935244). HBV infection is closely related to liver cancer(PMID:39578653). In the discussion part, we call for abdominal ultrasound, AFP and liver function examination for patients with HBsAg-positive patients. The clinical value of abdominal ultrasound is that it can evaluate the structural changes of the liver, such as liver cirrhosis and liver tumor. AFP is a sensitive indicator of liver cancer, and as its serum marker, the abnormal increase of AFP value is helpful for the early detection of liver cancer. Liver function tests help to understand the functional status of the liver and evaluate the degree of liver injury. These examinations can help monitor the progress of hepatitis B, find complications and guide treatment, see the 276-284 of the article and new references 43-45(PMID: 38583119, PMID: 39660335 and PMID: 36012671).
Round 2
Reviewer 1 Report
Comments and Suggestions for Authors
Issues are all addressed.
Reviewer 2 Report
Comments and Suggestions for Authors
None